# Characterizing the Changes in Permafrost Thickness across Tibetan Plateau

Yufeng Zhao [1], Yingying Yao [1,*], Huijun Jin [2,3], Bin Cao [4], Yue Hu [5], Youhua Ran [6] and Yihang Zhang [7]

1 Department of Earth and Environmental Sciences, School of Human Settlements and Civil Engineering, Xi'an Jiaotong University, Xi'an 710049, China
2 Northeast-China Observatory and Research Station of Permafrost Geological Environment (Ministry of Education), School of Civil Engineering, and Permafrost Institute, Northeast Forestry University, Harbin 150040, China
3 State Key Laboratory of Frozen Soils Engineering, Northwest Institute of Eco-Environment and Resources, Chinese Academy of Sciences, Lanzhou 730000, China
4 National Tibetan Plateau Data Center (TPDC), State Key Laboratory of Tibetan Plateau Earth System, Environment and Resources (TPESER), Institute of Tibetan Plateau Research, Chinese Academy of Sciences, Beijing 100045, China
5 State Key Laboratory of Geohazard Prevention and Geoenvironment Protection, Chengdu University of Technology, Chengdu 610059, China
6 Heihe Remote Sensing Experimental Research Station, Northwest Institute of Eco-Environment and Resources, Chinese Academy of Sciences, Lanzhou 730030, China
7 School of Humanities and Social Science, Xi'an Jiaotong University, Xi'an 710049, China
* Correspondence: yaoyy27@xjtu.edu.cn

**Abstract:** Permafrost impacts the subsurface hydrology and determines the transport of buried biochemical substances. Current evaluations of permafrost mostly focus on the overlying active layer. However, the basic but missing information of permafrost thickness constrains the quantification of trends and effects of permafrost degradation on subsurface hydrological processes. Our study quantified the long-term variations in permafrost thickness on the Tibetan Plateau (TP) between 1851 and 2100 based on layered soil temperatures calculated from eight earth system models (ESMs) of Coupled Model Intercomparison Project (the sixth phase) and validated by field observations and previous permafrost pattern from remote sensing. The calculated permafrost distribution based on ESMs was validated by the pattern derived from the MODIS datasets and field survey. Our results show that permafrost thicker than 10 m covers approximately 0.97 million km² of the total area of the TP, which represents an areal extent of over 36.49% of the whole TP. The mean permafrost thickness of the TP was 43.20 m between 1851 and 2014, and it would decrease at an average rate of 9.42, 14.99, 18.78, and 20.75 cm per year under scenarios SSP126, SSP245, SSP370, and SSP585 from 2015 to 2100, respectively. The permafrost thickness will decrease by over 50 cm per year in Qiangtang Basin under SSP585. Our study provides new insights for spatiotemporal changes in permafrost thickness and a basic dataset combined results of remote sensing, field measurements for further exploring relevant hydrological, geomorphic processes and biogeochemical cycles in the plateau cryospheric environment.

**Keywords:** permafrost thickness; Tibetan Plateau; soil temperature; MODIS; CMIP6

## 1. Introduction

Approximately $22.79 \times 10^6$ km² (21–23 million km²) of the exposed land area of the Northern Hemisphere is covered by permafrost [1–3]. In addition to its role in hydrological and biogeochemical processes [4,5], permafrost also triggers local geo-hazards, such as ground settlement and surface subsidence, that threaten the engineered infrastructure [6]. Due to a rapid rate of climate change, permafrost has been degrading extensively [7], potentially affecting the northern and alpine ecosystems, as well as the socioeconomic

system [8,9]. Thus, it is crucial to project the extent and rate of permafrost degradation over a long period of time, which will serve as a baseline for other relevant evaluations, including hydrological, biogeochemical, and socioeconomic processes. Permafrost degradation studies primarily focus on changes in the active layer subject to annual thawing and freezing.

The active layer interacts directly with the atmosphere and land surface; thereby its thickness and change can be determined via ground temperature measurement or numerical simulation [10–12]. The thickness of permafrost can be determined by ground temperature measured in boreholes in TP [13–15]. However, boreholes are often unable to penetrate through the permafrost layer (s) given its high cost, and fail to reveal spatial pattern. While the evaluation of changes in the active layer can provide valuable information regarding changes in surface waters and distribution of near-surface ground ice, the mechanisms behind these changes are often far more sophisticated than can be identified through these ground-based investigations [16]. As permafrost thaws, new pathways are becoming available for groundwater to flow with energy and solutes [17,18]. To make progress in predicting future climate change, it is necessary to integrate subsurface processes across at least the entire depth of the permafrost layer (s).

For assessing regional permafrost degradation, models have been developed in conjunction with spatial attribute datasets (e.g., landcover and hydrogeology) [19]. The approaches also concentrate on the near-surface ground thermal regime, and they require sufficient field measurements or using remote sensing products to interpolate a regional change [20]. For instance, the TTOP model aims to calculate the temperature at the top of permafrost [21,22]; GIPL2 model, which is a transient state numerical model to simulate the soil temperature while considering the phase changes between the ice and unfrozen water, is mostly applied to simulate the active layer thickness (ALT) [23]. The earth system models (ESMs) are essential tools for predicting and understanding regional and global changes in soil components [24]. Compared with the results from TTOP and other statistical models, which assume equilibrium conditions, the soil temperature results from CMIP6 are transient and physical. In the Climate Model Intercomparison Project 5 (CMIP5), ESMs are able to simulate soil heat transport and storage, because soil temperature is closely related to carbon storage and soil respiration [25]. ESMs of CMIP6 (latest generation) simulate soil temperature by considering snow cover and insulation, and the maximum depth of the model domain can reach 42 m [26,27]. Based on the simulated soil temperature, it is possible to map the regional degradation of permafrost at various depths.

The Tibet Plateau (TP), also known as world's third pole and one of the key Asia water towers, containing the largest permafrost region in the low–middle latitudes. About 60% of the TP is occupied by permafrost region [28,29]. Permafrost degradation has been documented from soil profiles in recent decades, resulting in hydrological changes, such as increased winter baseflow and thermokarst lakes [30]. The permafrost distribution and classification for TP have been studied since the early 1960s [31], and the temperature and thickness of permafrost have been extensively and continuously measured along the Qinghai–Tibet highway and railway for at least two decades [13]. Additionally, thermal indices or ALT were used to determine the changing extent, but long-term changes in permafrost thickness (PT) have not yet been systematically quantified.

To close this knowledge gap, we employ eight ESMs from CMIP6 to extract the layered soil temperature for the latest historic period (1851–2014) and the near future (2015–2100). Field observations and permafrost distribution derived from various remote sensing products were used to calibrate the simulated temperature generated by ESMs. Afterwards, the PT was determined using the geothermal gradient equation.

## 2. Materials and Methods

### 2.1. Permafrost Thickness Calculation

In this study, the geothermal gradient equation was used to calculate PT. Figure 1 illustrates the temperature changes along the vertical profile, the ALT, and the PT. The active

layer is subject to thawing and freezing throughout the year. The temperature beneath the ALT is generally below 0 °C in case of the attached permafrost and the permafrost soil is perennially frozen. The annual variation in temperature dampens exponentially with increasing depth, and below a certain level, the temperature remains largely constant throughout a year. A depth of zero annual amplitude ($DZAA$) is defined as the point at which the annual variation in temperature is undetectable (i.e., less than 0.1 °C in most cases). Since the changes in mean annual ground temperature (MAGT) change below the $DZAA$ are almost linear, we can calculate the PT as follows:

$$PT = \frac{T_{DZAA} - T_f}{G_T} + DZAA - ALT \tag{1}$$

where $T_{DZAA}$ is the MAGT at the $DZAA$, $T_f$ is the temperature of the bottom the permafrost (we set it as 0 °C), $G_T$ is geothermal gradient which is estimated using the inverse distance weighting (IDW) method from the observed $G_T$, which is identified by temperature from observed boreholes [32] and all data were provided in the Supplementary Materials, and $ALT$ is the active layer thickness [13].

### 2.2. Data of Soil Temperature from CMIP6 Projection

The study area bounded by longitudes 73°–104.5°E and latitudes 25°–40°N. The information of the study area is shown in Figure 2. The landscape was characterized by the Space Shuttle Radar Terrain Mission DEM data. The geographical attributes include the spatial extent of lakes and river basins [33]. The PT was estimated for each earth system model from CMIP6 and re-sampled to the same spatial resolutions of 0.5° using bilinear interpolation method. All results were then calculated as the ensemble mean of all ESMs.

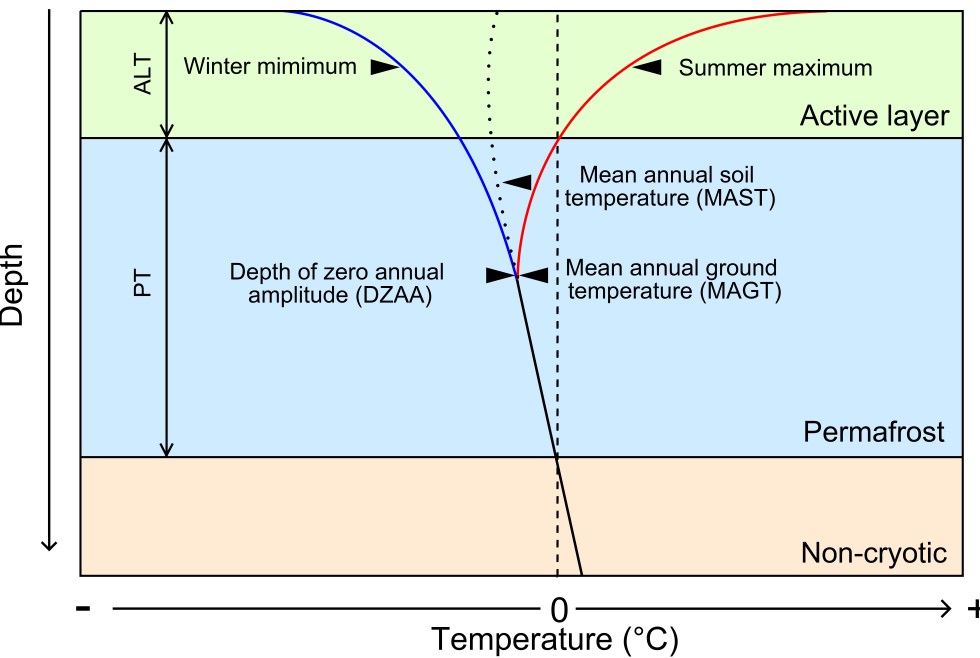

**Figure 1.** A typical ground temperature profile to illustrate permafrost thickness (PT) and active layer thickness (ALT) [34].

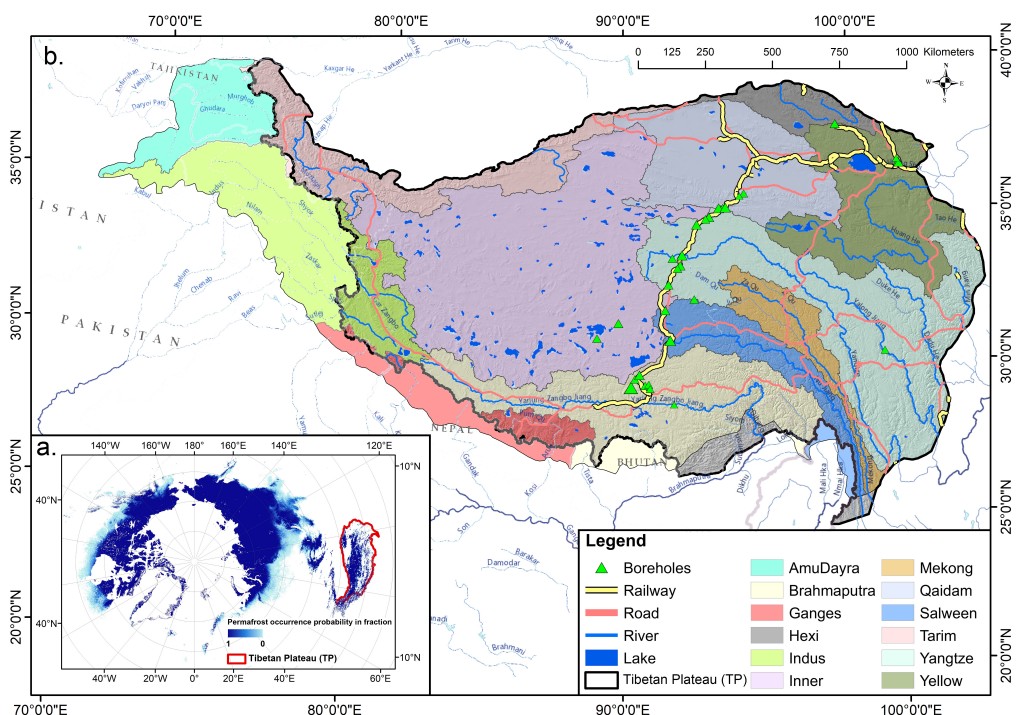

**Figure 2.** Study area. (**a**). The distribution of permafrost for the Northern Hemisphere [3], where the Tibetan Plateau (TP) is marked with a red margin. (**b**). The extents of TP and major river basins originating from the TP [35]. Boreholes in the map was used to measure ground temperatures along the Qinghai–Tibet railway and highway from Golmud, Qinghai Province to Lhasa, Tibet Autonomous Region, China. Redish lines of road networks indicate the national highways.

In this study, eight ESMs from CMIP6 with relatively deeper soil profile (i.e., depth > 30 m) were selected. The time resolution of all eight models is monthly and the details of eight CMIP6 models are summarized in Table 1. The models include a simulated historical dataset from 1851 to 2014 and four typical scenarios (SSP126, SSP245, SSP370, and SSP585) in the near future period of 2015 to 2100, which are combined with different shared socioeconomic pathways (SSP) and representative concentration pathway (RCP). The observed soil temperature data was obtained from published data [36], and was used to validate the simulated soil temperature by ESMs. The multiple mapped permafrost pattern [22,28,37,38] from MODIS were used to validate the calculated distribution of permafrost.

**Table 1.** Summary of the eight models from CMIP6.

| Name | Institute | Spatial Resolution (Longitude × Latitude) | Soil Depth (m) | Soil Layers |
|---|---|---|---|---|
| CESM2 | National Center for Atmospheric Research, USA | 1.25° × 0.94° | 42.0 | 24 |
| CESM2-WACCM | National Center for Atmospheric Research, USA | 1.25° × 0.94° | 42.0 | 24 |
| CMCC-CM2-SR5 | Fondazione Centro Euro-Mediterraneo sui Cambiamenti Climatici, Italy | 1.25° × 0.94° | 35.2 | 14 |
| CMCC-ESM2 | Fondazione Centro Euro-Mediterraneo sui Cambiamenti Climatici, Italy | 1.25° × 0.94° | 35.2 | 14 |
| FGOALS-f3 | Chinese Academy of Sciences, China | 1.25° × 0.9° | 35.2 | 14 |
| FGOALS-g3 | Chinese Academy of Sciences, China | 2° × 2.26° | 35.2 | 14 |
| NorESM2-LM | NorESM Climate modeling Consortium consisting of CICERO, Norway | 2.5° × 1.89° | 42.0 | 24 |
| NorESM2-MM | NorESM Climate modeling Consortium consisting of CICERO, Norway | 1.25° × 0.94° | 42.0 | 24 |

## 3. Results

### 3.1. Validation of Soil Temperature Simulations

The soil temperatures at different depths simulated by ESMs were validated by observations and other independent permafrost study results across TP. The comparison shows that the performance of the simulation of ESMs for the TP is generally satisfactory, with a mean value of root mean square error (RMSE) over 4.64 shown in Figure 3. The relatively

better-fitting ESMs are CMCC-CM2-SR5 and CMCC-ESM2 with RMSE values over 3.33 and 3.26, respectively. To validate the results of spatial distribution of permafrost calculated from ESMs, we further counted the grids of permafrost coverage based on soil temperature and compared it with the previous results derived from remote sensing products. Since eight ESMs have different resolutions from $2.5° \times 1.89°$ to $1.25° \times 0.94°$; the total number of grids of eight ESMs are between 270 and 1015. In counting the permafrost coverage area, we performed a probability calculation. We used data from 2000 to 2014, and for a certain grid, the probability of permafrost occurrence is $(m \times n)/(15 \times 8)$, if m years and n ESMs have permafrost. Figure 4 shows the calculated permafrost distribution based on ESMs keep consistency with the previous results, particular close to the map derived from MODIS-LST remote sensing product by Obu et al. [37]. Therefore, the soil temperature simulated by ESMs can be a reliable dataset to estimate the thickness of permafrost.

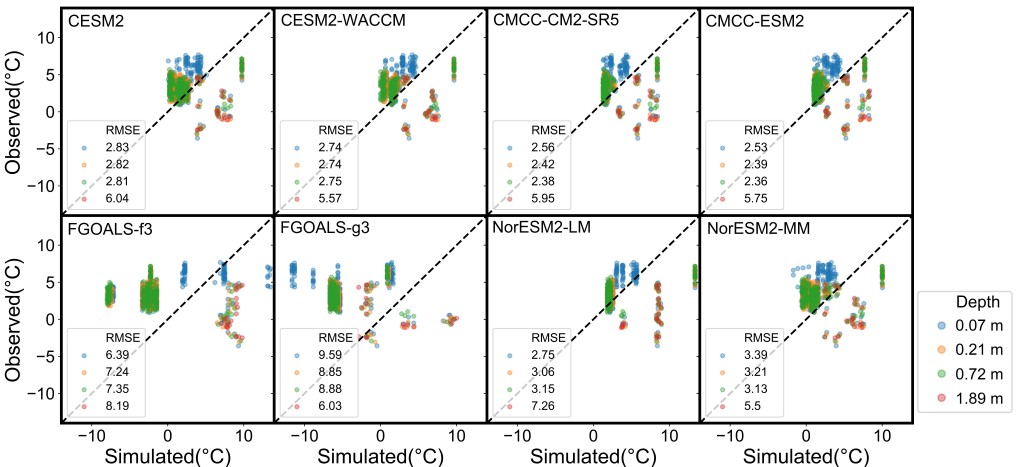

**Figure 3.** Comparison of simulated soil temperatures from eight ESMs and observed data at different depths for the TP.

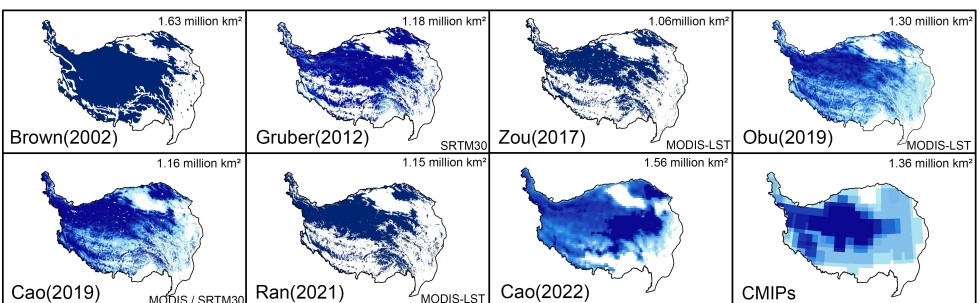

**Figure 4.** Spatial comparison of permafrost distribution from previous results and our results. The results of Brown et al. [39], Zou et al. [22], and Ran et al. [38] directly show the extent of spatial permafrost, while others represent the permafrost distribution using the possibility of permafrost coverage represented as shades of color [1,27,28,37].

### 3.2. Evaluation for Spatial Distribution of Permafrost Thickness

We calculated the permafrost thickness of TP from 1851 to 2014, which is the historical period assigned by CMIP6 models. Figure 5 shows the spatial distribution of permafrost thickness for every 20 years. The total area of permafrost with thickness over 10 m is approximately 0.97 million km², and covers 36.49% of the total area of the TP. This area decreased by 7.21% during the historical period from 1851 to 2014. The thick permafrost (>40 m) was mainly distributed in the interior basins and high-mountains and high-plateaus and the upper reaches of Yangtze and Yellow river basins. Thick permafrost accounts for an area of 0.48 million km², and decreaseds by about 27.08% from 1851 to 2014. Permafrost-free areas

are mainly distributed in the northern Qaidam Basin and on the eastern and southeastern TP, and their areal extents barely changed. The permafrost on the southwestern TP began to disappear from about 1931 to 1950.

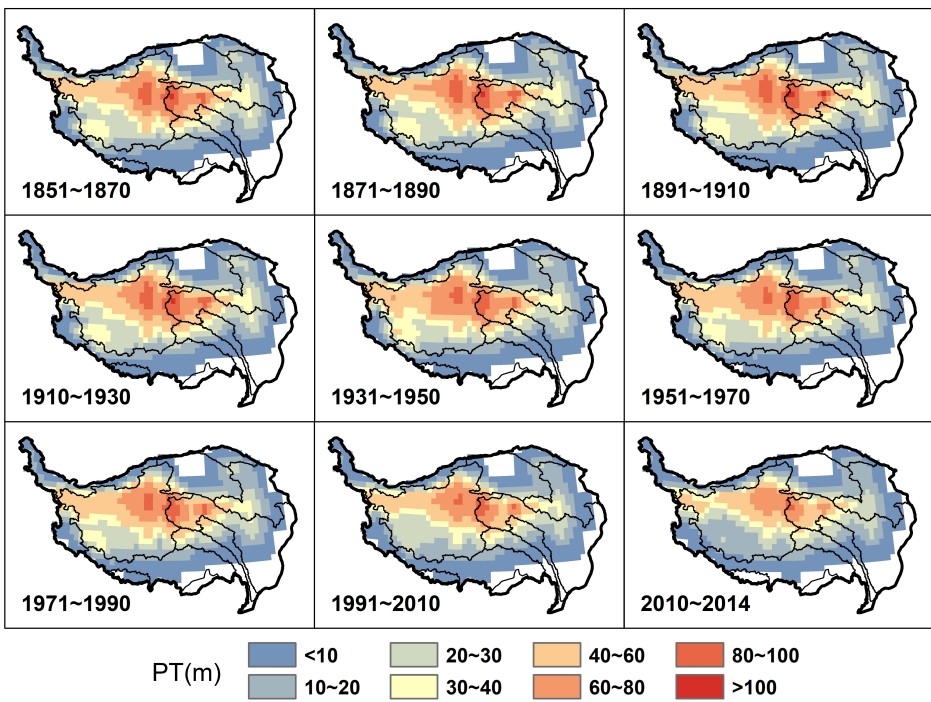

**Figure 5.** Spatial distribution of model-rebuilt permafrost thickness on the Tibetan Plateau, Southwest China during 1851–2014.

We calculated the permafrost thickness of TP from 2015 to 2100 and the average for every 20 years using the future data set of ESMs from CMIP6, including four different scenarios (SSP126, SSP245, SSP370, and SSP585). Figure 6 shows the spatial distribution of computed permafrost thickness. In the near future, permafrost thickness will have an apparent decreasing trend under the four scenarios. The minimum and maximum reductions in areal extents of permafrost thicker than 10 m will be reduced to 0.19 and 0.87 million km$^2$ by 2100 under scenarios SSP126 and SSP585, respectively. The area of permafrost thicker than 30 m will reduce from 0.55 million km$^2$ in 2014 to 0.18, 0.006, 0, and 0 million km$^2$ in 2100 under scenarios SSP126, SSP245, SSP370, and SSP585, respectively.

### 3.3. Future Reduction Trends of Permafrost Thickness

Figure 7 shows average variations in permafrost thickness between 1851 and 2100. The average permafrost thickness was 43.20 m during historical period of 1851–2014, and by 2100 it would be reduced to 22.17, 14.02, 7.07, and 5.67 m under scenarios SSP126, SSP245, SSP375, and SSP585, respectively. The time series of PT indicates a relatively stable permafrost during the historical period before 1990. However, it has entered a period of rapid permafrost degradation since 2008.

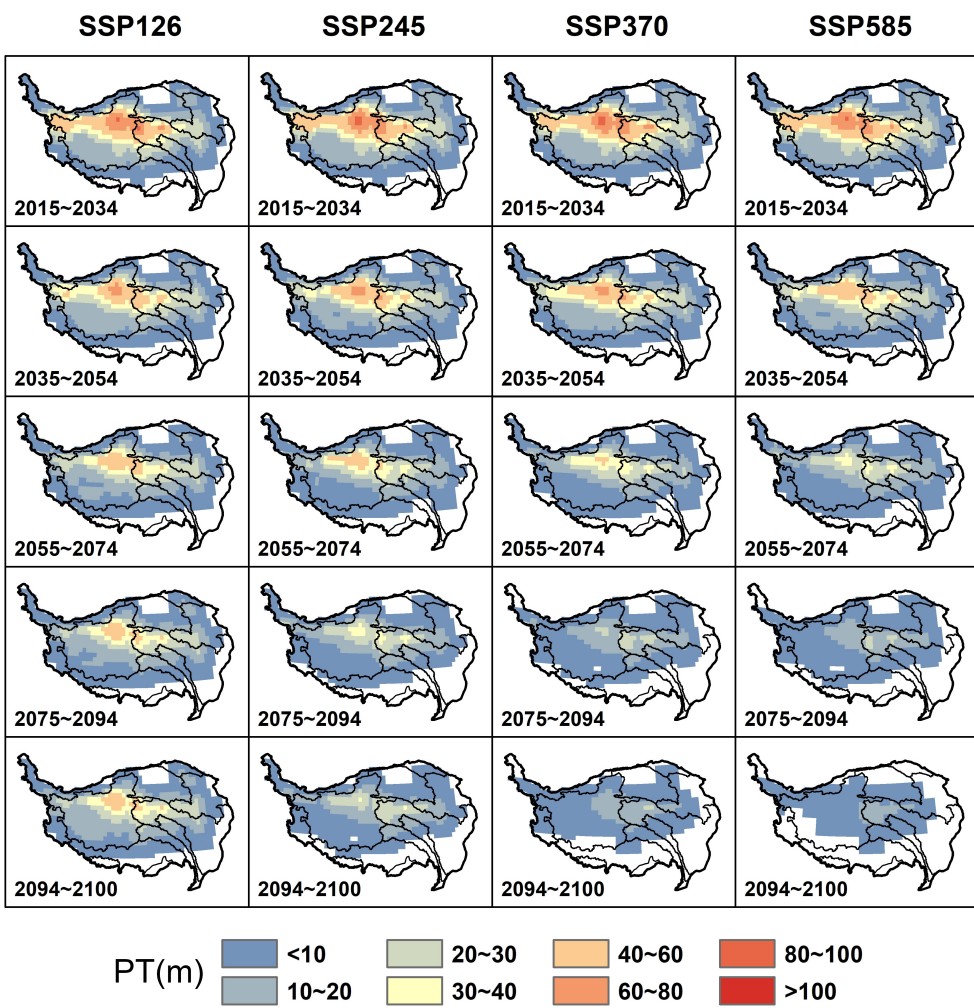

**Figure 6.** The Earth Systems Models (ESMs) predicted spatial distribution of plateau permafrost thickness in the near future (2015–2100).

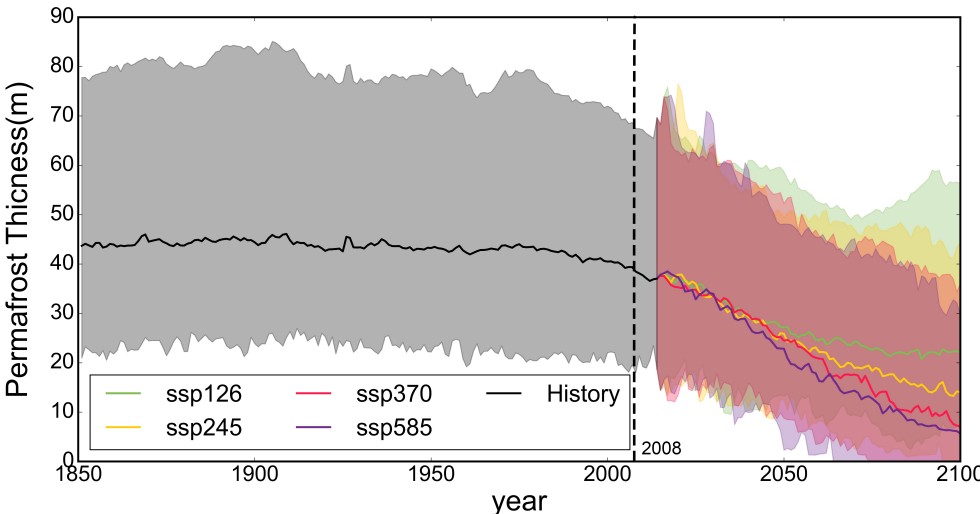

**Figure 7.** Temporal trend of average thickness of permafrost between 1851 and 2100 computed by eight ESMs across the Tibetan Plateau in Southwest China. The gray and color ranges represent the maximum and minimum ranges predicted from the ESMs under four scenarios.

The spatial distribution of linear trends of permafrost thinning were calculated and shown in Figure 8. The average PT for the entire TP would decrease by about 9.42, 14.99, 18.78, and 20.75 cm per year under scenarios SSP126, SSP245, SSP370, and SSP585 from 2015 to 2100, respectively. The area with an annual thinning rate or more than 50 cm, accounts for 12.39% of the total permafrost area under SSP585, and it mainly occurs on the interior TP. In addition, the average trends for ten major river basins on the TP were counted (Figure 8 and Table 2). The area of central TP which is marked as Inner (Qiangtang) Basin in this study, Tarim and Yangtze river basins are top three river basins with the greatest reduction in PT. Especially for the Inner (Qiangtang) basin, the average PT of this area will decrease from 33.54 to 19.22 m even under a sustainability scenario (SSP126).

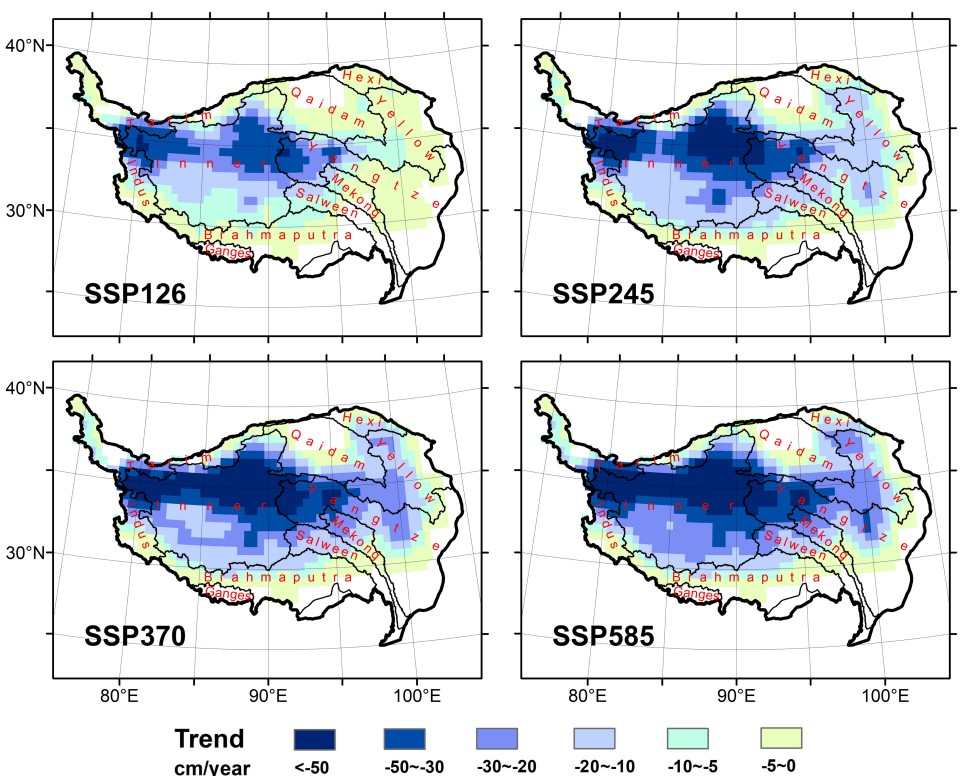

**Figure 8.** Spatial distribution of linearly regressed thinning trends in permafrost thickness across the Tibetan Plateau during 2015–2100 as model predicted under the four scenarios of SSP125, SSP245, SSP370, and SSP585.

**Table 2.** The averaged linear trends (cm/year) in permafrost thickness for major river basins across Tibetan Plateau.

| Basin | SSP126 | SSP245 | SSP370 | SSP585 |
|---|---|---|---|---|
| TP | −9.42 | −14.99 | −18.78 | −20.75 |
| Brahmaputra | −2.15 | −3.57 | −4.06 | −4.51 |
| Hexi | −1.99 | −4.7 | −6.65 | −8.15 |
| Indus | −11.96 | −13.54 | −14.79 | −17.5 |
| Inner | −20.53 | −31.43 | −38.63 | −42.62 |
| Mekong | −3.49 | −9.12 | −12.91 | −13.29 |
| Qaidam | −2.92 | −6.24 | −9.1 | −10.14 |
| Salween | −4.84 | −7.83 | −10.53 | −11.31 |
| Tarim | −9.4 | −10.82 | −12.44 | −16.02 |
| Yangtze | −8.31 | −14.15 | −18.06 | −18.58 |
| Yellow | −3.87 | −9.75 | −13.54 | −15.42 |

## 4. Discussion

Based on the model computed soil temperatures from eight ESMs of CMIPs, our study not only provides the spatial distributed data of permafrost thickness, but also those of the thinning trend of permafrost in the near future (2015–2100). When compared with previous estimates of Wu et al. [13] based on field observations along the Qinghai-Tibet railway, the calculated PT are generally better with a mean deviation of 12.20 m. Certain points are subject to large errors due to the regional nature of our permafrost thickness estimate. In this regard, there are still unavoidable differences with the regional estimations and local observations. Even though the datasets directly from ESMs have coarse spatial resolutions, they are capable of providing a preliminary estimate of the multi-year thickness of permafrost, which is an unvisitable and difficult quantity to measure. Remote sensing of surface temperatures can be used to validate CMIP6 simulated results and quantify the data discrepancies at the regional scale to further gain an understanding of subsurface conditions. As roughly estimated in previous studies, the permafrost thickness across TP is greatest in high mountainous areas, up to 200–400 m in upland areas, followed by that of 60–130 m the hilly areas between Kunlun and Tanggula mountains, and the least (0–60 m) in high plains and river valleys [40]. According to our results, permafrost thickness distributions under various elevation topography conditions are more accurately quantified. In areas with elevations over 4000 m a.s.l. in TP, the permafrost thickness varies between 0.03 and 84.77 m, whereas in river valley areas, it varies between 0 and 68.04 m during historical period of 1851–2014. While our estimate of permafrost extent (1.36 million $km^2$) is within the average range (1.06–1.63 million $km^2$), the permafrost area may reach two to three times of its current estimation during the Last Ice Age (i.e., Last Glaciation Maximum (LGM) or Last Permafrost Maximum (LPMax) at 26–19 ka BP) [41]. Our estimation of permafrost thickness is based on an assumption of top–down thaw processes which are mainly characterized as the thickening of the active layer and formation of thawing interlayers. However, permafrost degradation also involves bottom–up and lateral thaw processes that occur in talik or active layer zones and are facilitated by lateral and preferential flow [42]. These processes may result in rapid permafrost degradation (abrupt permafrost degradation), causing our estimations to be conservative. As a result of successive climate changes, multiple layers of permafrost (i.e., thaw interlayer) may form during permafrost degradation, and these layers have been extensively identified on the eastern side of the Qinghai–Tibet Railway [43–45]. This is not taken into account by our top–down estimation method, so there may be some local inconsistencies. Moreover, there are a number of other factors that influence permafrost degradation, which can also impair our estimates of permafrost thickness. As snow cover has an insulating effect, it limits the loss of winter heat from the ground and minimizes the impact of changes in the air temperature on the ground thermal regime [46]. Vegetation canopy reduces solar radiation reaching the ground surface, thus lowering the ground temperature [47]. Therefore, the ESMs need to take these surficial factors into account in order to improve the accuracy of soil temperature simulations.

By analyzing the thinning trend of permafrost in different watersheds, our study provides a basis for determining the effect of permafrost on hydrological processes. On the TP, the area with the greatest permafrost thinning is the Inner (i.e., also called as the Qiangtang Plateau) Basin with 29.00 cm/year precipitation and 21.05 cm/year evaporation [48–50]. Increasing ALT and decreasing PT will further intensify surface water infiltration and alter the original water cycle [51,52]. For the outflow basins surrounding the Qiangtang Plateau, including the Yangtze, Yellow, and Tarim rivers basins, permafrost degradation is primarily observed in the headwater areas. Permafrost degradation may mainly affect their source areas. In the Yellow River Basin, for example, over 87% of the headwater region is covered by permafrost, and its degradation has had an effect on run-off of the upper Yellow River [53] . Although permafrost degradation has a small proportional impact on the total river discharges in mid- and down-streams, it can alter upstream ecohydrological processes.

Permafrost thickness quantification provides important basic data for an in-depth study of permafrost degradation on ground ice estimation, groundwater flow, shaping of thermokarst lakes and drained basins, and further quantification of permafrost carbon release. During permafrost thaw, ground ice melting may generate water flow; Previous studies have estimated a storage of 9528 km$^3$ of ground ice at the top 1–10 m in depth in plateau permafrost regions [54]. This value may be underestimated in light of our average permafrost thickness of 43 m across TP. In terms of subsurface system, the permafrost layer can serve as a barrier. As the permafrost layer thins, the barrier's role diminishes, and the average hydraulic conductivity increases, permitting a gradual transition to a semi-aquifer or even aquifer [55]. The opened aquifer can change the circulation and material transport path of the whole groundwater flow system [56]. As a result of our research, we have been able to provide important thickness data for developing groundwater system models. One of the most significant features of permafrost degradation on the Tibetan Plateau is the formation of thermokarst lakes and drained basins. In the event that talik forms, groundwater flow may serve as a reliable source of water for recharging thermokarst lakes [42]. Based on the thickness of the permafrost, talik formation can be predicted not only for pattern mapped from remote sensing but considering subsurface flow mechanism [57]. In grassland ecosystems on the Tibetan Plateau, soil organic carbon stocks at 2 m depth in the permafrost zone are estimated to be around 28 Pg, while soil organic carbon stocks at 0–25 m depth may be 160 Pg [58,59]. Although deep carbon stocks are smaller than shallow carbon stocks, more accurate information on permafrost thickness can provide better estimates of carbon stocks.

## 5. Conclusions

This study evaluates the spatial distribution and changes of permafrost thickness across the TP based on the layered soil temperatures predicted from the eight ESMs from CMIP6. The simulated soil temperature datasets were validated through field observations and MODIS-based pattern. Furthermore, long-term changes in permafrost features were quantified to identify the trends of permafrost degradation. Major findings include the following:

(a) The simulated errors of layered soil temperatures by ESMs of CMIP6 for alpine regions are within the acceptable range, while the CMCC-CM2-SR5 and CMCC-ESM2 model results are better on the TP. The permafrost distribution and calculated permafrost extents (1.36 million km$^2$) based on ESMs of CMIP6 are consistent with other recent evaluations (1.06 to 1.63 million km$^2$).

(b) The total area of permafrost exceeding 10 m in thickness is approximately 0.97 million km$^2$, which represents over 36.49% of the total area of the TP. Permafrost thickness exceeding 40 m were mostly found in the Inner (Qiangtang) Basin, in the headwater areas of the Yangtze and Yellow rivers.

(c) During the historical period of 1851–2014, the average permafrost thickness was 43.20 m. By 2100, this thickness would be reduced to 22.17, 14.02, 7.07, and 5.67 m under climate change scenarios SSP126, SSP245, SSP375, and SSP585, respectively. There are 12.39% of the total permafrost area under the scenario SSP585 with an annual downward thinning rate of permafrost over 50 cm, mainly found on the interior TP.

Our study provides a direct and comprehensive dataset for analyzing the distribution, development, and degradation of permafrost across the TP. These more accurate data on permafrost thickness can greatly help to further explore the mechanisms of subsurface hydrological processes, to identify thermokarst lakes and drained basins, and to evaluate permafrost carbon stocks and release potential.

**Supplementary Materials:** The following supporting information can be downloaded at: https://www.mdpi.com/article/10.3390/rs15010206/s1, Table S1: Geothermal gradient ($G_T$) used in this study.

**Author Contributions:** Conceptualization, Y.Y. and B.C.; methodology, Y.Z. (Yufeng Zhao), Y.Y. and B.C.; code, Y.Z. (Yufeng Zhao); resources, Y.Y.; writing—original draft preparation, Y.Z. (Yufeng Zhao), Y.Y. and H.J.; writing—review and editing, Y.Z. (Yufeng Zhao), Y.Y., H.J., B.C., Y.H., Y.R. and Y.Z. (Yihang Zhang); funding acquisition, Y.Y. All authors have read and agreed to the published version of the manuscript.

**Funding:** This study was supported by the Strategic Priority Research Program of the Chinese Academy of Sciences (grant number: XDA19070204), the National Natural Science Foundation of China (grant number: 41901023, 92047202), and the National Key R&D Program of China (Grant No. 2020YFC1808300).

**Data Availability Statement:** The CMIP6 dataset can be downloaded at https://esgf-node.llnl.gov/projects/cmip6/ (accessed on 31 October 2021). We collected soil temperature data from 78 sites located in permafrost regions from a variety of sources and datasets. It consists of eight sites from HiWA-TER [60], 10 sites from the Tibetan Plateau observatory of plateau scale soil moisture and soil temperature (Tibet-Obs) [61], and 60 sites from the multiscale Soil Moisture and Temperature Monitoring Network in the Central Tibetan Plateau (CTP-SMTMN) [62]. The HiWATER dataset is from the National Tibetan Plateau Data Center (https://doi.org/10.3972/hiwater.001.2019.db (accessed on 31 October 2021)), and Tibet-Obs and CTP-SMTMN are available from the National Tibetan Plateau Data Center (https://data.tpdc.ac.cn/zh-hans/data/ef949bb0-26d4-4cb6-acc2-3385413b91ee/ (accessed on 31 October 2021)).

**Conflicts of Interest:** The authors declare no conflicts of interest.

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
