# Peer review of "Characterizing the Changes in Permafrost Thickness across Tibetan Plateau"

_remotesensing, doi:10.3390/rs15010206_

Round 1
Reviewer 1 Report
This paper employs eight earth system models (ESMs) of Coupled Model Intercomparison Project to simulate the soil temperature data, which are compared with field observations. The simulated distribution of permafrost is also compared with remote sensing observations. Both show comparable results. The thickness of permafrost in Tibetan Plateau (TP) between 1851 and 2100 is calculated based on the simulated soil temperature, hence, the future trends of spatiotemporal changes in permafrost thickness can be provided. In general, the paper is interesting and worthy of discussion. The figures are well drawn, and the texts are clear. However, the corresponding method requires more elaborations.
1. In introduction, any related work that predicts the thickness of permafrost in other region? Which method has been used? What is the novelty of your method?
2. Is the geothermal gradient used in formula (1) a constant? Have you considered its variation with places?
3. The paper mentioned that the deviation of the simulated temperature is acceptable, in which criterion? Have you considered the uncertainties/error of the predicted thickness caused by the temperature deviation?
4. Why the calculated thickness value is not compared with the measurement along the Qinghai-Tibet highway and railway? How large is the deviation?
5. How does the simulated thickness value change in the scenario of multiple layers of permafrost?
6. Minor errors:
Line 23: remove “,” before [4,5]
Fig. 3: Please add the names of different models in the figure.
Line 124: close to the map derived from MODIS-LST remote sensing product , which one, three of them are marked as MODIS-LST.
Line 178-179, is varied ->” varies”
Author Response
We are very grateful for your time and efforts in helping us improve our manuscript. In response to your suggestions, we have revised the manuscript, with particular attention to the novelty of the method and the specific explanation of the geothermal gradient data. Further, our results of permafrost thickness have been compared with those measured along the Qinghai-Tibet highway and railway following your suggestion. A detailed point-by-point rebuttal that incorporates all responses to comments is attached.

Reviewer 2 Report
Review
The work is devoted to modeling spatiotemporal changes of the permafrost thickness in the Tibetan Plateau from 1851 to 2014 based on soil layer temperatures calculated from eight earth system models using remote sensing and field data. The authors quantified long-term changes in the characteristics of permafrost, on the basis of which the best models were identified, which are consistent with other recent estimates, and the area occupied by permafrost with a particular thickness was also calculated. An important part of the work is that the researchers have identified the average thickness of permafrost over the past periods from 1851 to 2014. and trends of change (thinning) of permafrost thickness in the near future are given.
The reviewer has the following suggestions and comments on the article:
1) In section 3.1. the researchers write that ESM-modeled soil temperatures at different depths have been validated by observations and other independent permafrost study results across Tibetan Plateau. What are these studies? The article should include references to the work.
2) The authors write that their calculations were confirmed by field data, but in the work these data are not displayed in any way, only the location of the wells is shown in Fig.2.
3) The authors for the past years (1851-2014) calculated the change in the thickness of permafrost, however, what factors caused this are not presented in the work. It would be necessary to provide data on air temperature, precipitation, snow cover, etc. Show this factors in the form of graphs and correlate these data with the change in the permafrost characteristic under consideration, explain such a change by the influence of factors.
4) It is not entirely clear why the authors conducted the study only until 2014, and the period from 2015 onwards was considered to be in the future? After all, those years have already passed. It was possible to calculate the future period, for example, from 2020 at least.
In general, the study is presented quite well, new interesting results are presented, the work can be recommended for publication if the article is supplemented with the above recommendations.

Author Response
We would like to extend our sincere thanks to you for your suggestions. We greatly benefit from it as it helps us to improve the manuscript. Each of your comments has been addressed and the manuscript has been revised accordingly. In particular, we would appreciate your comments regarding data clarification and results validation with previous studies. Attached is a detailed point-by-point response to all comments from reviewers.
